# Review of Vulnerability Factors Linking Climate Change and Conflict

**Takato Nagano and Takashi Sekiyama ***

Graduate School of Advanced Integrated Studies in Human Survivability, Kyoto University, Kyoto 606-8306, Japan; nagano.takato.83w@st.kyoto-u.ac.jp
* Correspondence: sekiyama.takashi.2e@kyoto-u.ac.jp

**Abstract:** This systematic literature review gathers societal vulnerability factors linking climate change and conflict from 53 existing studies. The findings reveal three main points. First, four relevant factors are missing from a previous vulnerability analysis framework proposed by Pearson and Newman: land degradation/land cover, gender, customs, and geographical conditions. Second, two factors, access to technology (e.g., for climate change adaptation) and partially democratic states, are insufficiently studied. Third, classification criteria in the previous framework need revision for accuracy. Considering these points, this study proposes a modified vulnerability analysis framework and offers five suggestions for future research directions in climate security research. First, more qualitative case studies are needed to complement the quantitative work. Second, in particular, cases where conflict was avoided or cooperation was established in high vulnerability areas need further research. Third, further research is needed on understudied factors (e.g., access to technology and partial democracy) and on factors the conventional framework cannot explain (e.g., land degradation/land cover, gender, customs, and geographical conditions). Fourth, no single vulnerability factor leads to conflict in isolation, but only in interaction; their connections must be studied. Finally, case studies are needed on vulnerability factors in countries and regions that have suffered from climate change but have not experienced conflict.

**Keywords:** climate change; conflict; security; vulnerability; systematic review





## 1. Introduction

The potential link between climate change and conflict has been the topic of increasing discussion in academia [1]. A large body of previous research has indicated that climate change may increase the risk of violent conflict [2–4]; however, it has also been mentioned that violent conflicts do not necessarily occur as a result of climate change [1,5–7]. In other words, opinions in academia are divided about the causal relationship between climate change and conflict. In addition, violent conflicts sparked by extreme weather events or natural disasters caused by climate change may or may not lead in turn to conflict in complex causal processes. Conflict is an extreme and rare consequence that occurs only under certain conditions and is not inevitable as a result of climate change. For example, several studies have pointed out that drought in the Fertile Crescent was a factor in social instability at the start of the Syrian Civil War [8,9]. However, it has also been pointed out that droughts that occurred during the same period did not lead to conflicts in neighboring countries such as Israel, Jordan, and Lebanon [1]. In other words, even if exposed to similar extreme weather events and natural disasters, some societies may be linked to conflict, whereas others may not.

Therefore, we may ask, what factors determine whether climate change leads to conflict? Why did the drought that occurred in the Fertile Crescent cause civil war in Syria, while it did not lead to civil war in neighboring countries? Even if a causal relationship exists between climate change and violent conflict, some researchers claim that there are

complex factors affecting that relationship, for example, social, economic, and political factors and so on [10–12]. With regard to the possible mechanisms linking climate change and conflict, the direct causal link between climate change itself (e.g., drought, global warming) and conflict has often been examined [13]. Until a few years ago, the mechanisms between climate change and conflict were rarely mentioned in the context of regional and national vulnerability (sensitivity, adaptive capacity, etc.) [13]. In particular, the general risk of violent conflict as an aspect of vulnerability has not yet been considered [13,14].

Sensitivity "is the degree to which a system is modified or affected by perturbations" [15]. Sensitivity is the condition that determines the extent to which a system or community is affected by climate change. Adaptive capacity is defined as the ability of a system to change to cope with the stress it faces due to its exposure and sensitivity [15]. In other words, adaptive capacity can be described as the ability of a community or system to control the (adverse) effects of climate change. The adaptive capacity of each society can vary depending on the institutions and customs of that society, and can vary depending on how good or bad governance is [16]. The level of social services, such as health insurance and education, also has a significant influence on the ability of socially vulnerable people (e.g., the elderly and children) to cope with the effects of climate change and, if necessary, find alternative means of coping [13]. Thus, in line with the Intergovernmental Panel on Climate Change (IPCC) definition, Schilling et al. classified adaptive capacity into two categories: general adaptive capacity, which is a general condition and indicator [17]; and specific adaptive capacity, which is an indicator specific to a certain impact [17]. General adaptive capacity can significantly reduce the vulnerability of personal income and human development [17], social services such as healthcare, and physical infrastructure such as irrigation systems and well-maintained buildings and roads [18]. Thus, the general adaptive capacity depends on the availability of social services, physical infrastructure, and the economic capacity to access these services and systems. Specific adaptive capacity, on the other hand, includes institutional performance, availability of knowledge and technology, and so on [17]. In other words, it is important to have the capacity, institutions, and habits to adequately provide and use the aforementioned physical infrastructure and social services. The general risk of violent conflict is defined as the likelihood of violent conflict breaking out in a certain area [14]. In general, countries with high population size, low per capita income, recent political instability and lack of established democratic institutions, small military forces and tortuous terrain, and low economic growth rates, and countries located in undemocratic regions, are highly correlated with the occurrence of civil war [19]. Furthermore, ethnic differences within groups have been found to be highly correlated with conflicts involving small arms [19]. On the other hand, in the context of the link between climate change and conflict, general risk factors for conflict include low economic growth, low levels of democratization, and past experience with conflict [14]. Moreover, some literature points out that ethnic divisions are one of the main determinants of the risk of armed conflict [20–23]. Although civil wars are not necessarily rooted in ethnic tensions, almost two-thirds of civil wars since 1946 have been fought on ethnic lines [24].

Pearson and Newman is one of the few review articles on social vulnerability linking climate change and conflict [13]. They proposed to understand vulnerability in terms of three aspects: sensitivity, adaptive capacity, and general risks of violent conflict. However, given their focus on the African agricultural sector, vulnerability in other contexts is not necessarily within their scope. It is also not clear which factors are included in these three aspects.

Therefore, this study uses a systematic literature review approach to identify the factors identified in previous studies as vulnerabilities in society that link climate change and conflict. Studies included in the review were those that mentioned vulnerability factors linking climate change and conflict from the perspectives of sensitivity, adaptive capacity, and general risks for violent conflict. The fields of the reviewed studies covered diverse fields such as natural resources, agriculture, economic and political systems. Based on the results of this review, this study proposes a modified vulnerability analysis framework.

This analytical framework provides clues regarding the factors that should be focused on when studying the mechanisms linking climate change and conflict.

## 2. Materials and Methods

### 2.1. The Criteria for Studies in the Review

Climate security research has increased in academia since around 2007 [25–27]. Hence, this research included studies published between 1 January 2007, and 23 July 2022. The key search terms were climate, conflict, and vulnerability. These terms were searched in two electronic scholarly databases, Scopus and Web of Science (WoS).

The criteria for studies included in the review were those that mentioned vulnerability factors linking climate change and conflict. The screening procedure is presented in Table 1 and Figure 1. As a result of this research, 39 studies were found in Scopus and 36 studies in Web of Science. From these, duplicates were excluded. This systematic literature review included 53 studies; 24 out of the 53 quantitative studies, 13 out of the 53 studies were qualitative, 12 out of the 53 studies were reviews, and 4 out of the 53 studies were mixed-methods studies, which included both quantitative and qualitative aspects (see Appendix A).

**Table 1.** Keyword search flow for studies.

| Scopus | |
|---|---|
| Search Query | Climate & Conflict & Vulnerability |
| Total number of studies | $n = 441$ |
| (1) Screening from title | $n = 104$ |
| (2) Screening from abstract | $n = 40$ |
| (3) Screening from introduction | $n = 39$ |
| Web of Science | |
| Search Query | Climate & Conflict & Vulnerability |
| Total number of studies | $n = 386$ |
| (1) Screening from title | $n = 135$ |
| (2) Screening from abstract | $n = 47$ |
| (3) Screening from introduction | $n = 36$ |

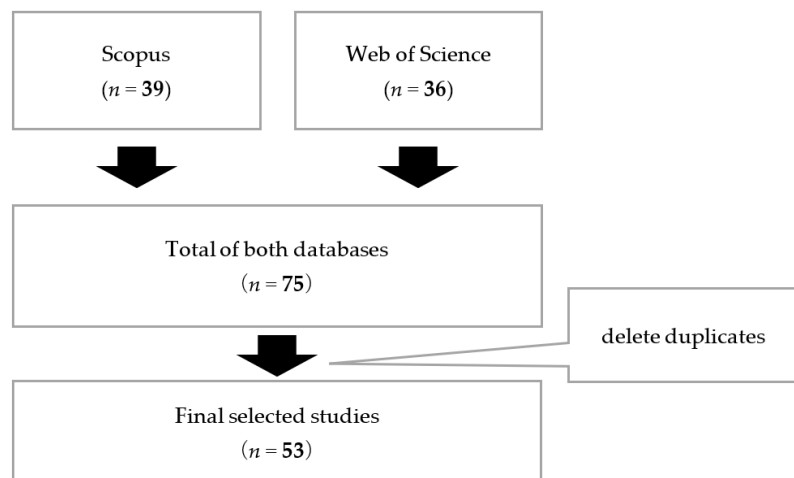

**Figure 1.** Studies screening flow.

### 2.2. Classification

This research summarizes the factors pointed out in previous studies by referring to Pearson and Newman's (2019) classification. They classified vulnerability factors into a total of 11 types under three categories (sensitivity, adaptive capacity, and general risks for violent conflict). This study first classified the vulnerability factors identified in the 53 previous studies into these 11 categories. Then, factors that could not be classified into

the 11 categories were classified into 4 additional categories. In addition, as explained below in the Discussion section, some Pearson and Newman's criteria for classifying each vulnerability factor was reconsidered through the literature review process. Specifically, "Low levels of development" was moved from the "Genera risks for violent conflict" category to that of "Adaptive capacity". Also, "Political and ethnic marginalization" was moved to the "General risks for violent conflict" category. The following Table 2 shows the classification of vulnerability factors used in this study.

**Table 2.** Classification of vulnerability factors. The factors in red show the differences from the previous framework proposed by Pearson and Newman.

| Factors | Pearson and Newman's (2019) Classification | This Research Classification |
|---|---|---|
| (1) Dependence on and access to natural resources | ○Sensitivity | ○Sensitivity |
| (2) Dependence on agriculture | ○Sensitivity | ○Sensitivity |
| (3) Land degradation/land cover | × | ○Sensitivity |
| (4) Gender | × | ○Sensitivity |
| (5) Better governance | ○Adaptive capacity | ○Adaptive capacity |
| (6) Development of economic system | ○Adaptive capacity | ○Adaptive capacity |
| (7) Access to technology | ○Adaptive capacity | ○Adaptive capacity |
| (8) Customs | × | ○Adaptive capacity |
| (9) Low level of development | ○General risks for violent conflict | ○Adaptive capacity |
| (10) Low levels of economic growth | ○General risks for violent conflict | ○General risks for violent conflict |
| (11) Partial levels of democracy | ○General risks for violent conflict | ○General risks for violent conflict |
| (12) High population | ○General risks for violent conflict | ○General risks for violent conflict |
| (13) Recent conflict and tensions | ○General risks for violent conflict | ○General risks for violent conflict |
| (14) Reduce political and ethnic marginalization | ○Adaptive capacity | ○General risks for violent conflict |
| (15) Geographical conditions | × | ○General risks for violent conflict |

## 3. Results

Based on the results of this review, this section first provides an overview of previous studies on vulnerability. Second, this research summarizes the factors pointed out in previous studies.

### 3.1. Overview of Previous Research on Vulnerability

The vulnerability factors identified in the 53 studies reviewed can be summarized into 15 types (Figure 2). Among the total number of quantitative, qualitative, review, and mixed studies, vulnerabilities with the highest number of mentions were, from the top, Better governance (33), High population (28), Dependence on and access to natural resources (23), Recent conflicts and tensions (20), Reduced political and ethnic marginalization (19), and Dependence on agriculture (19). On the other hand, factors mentioned in few studies

are Access to technology (1), Gender (3), Land degradation/land cover (7), Customs (8), Geographical conditions (8), and Partial democracy (11).

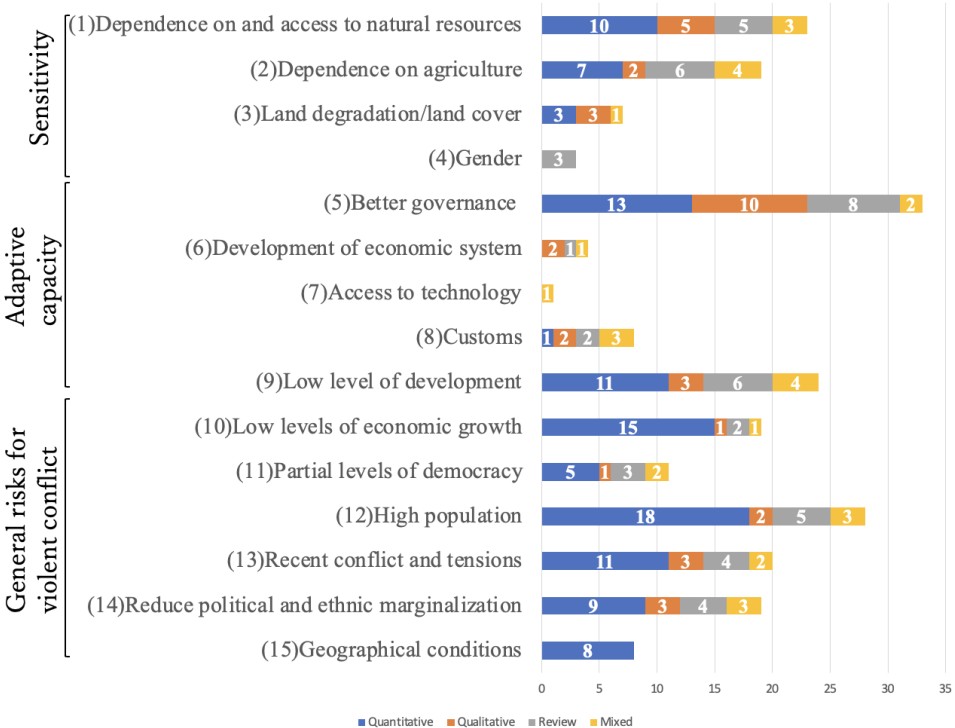

**Figure 2.** Overview of previous research on vulnerability.

In quantitative studies, vulnerabilities with the highest number of mentions were, from the top, High population (18), Low levels of economic growth (15), Recent conflicts and tensions (11), and Better governance (13). On the other hand, vulnerability factors with particularly few mentions are, from the bottom, Gender (0 mentions), Development of the economic system (0), Access to technology (0), Customs (1), and Land degradation/land cover (3).

In the total number of qualitative studies, vulnerabilities with the highest number of mentions were, from the top, Better governance (10) and Dependence on and access to natural resources (5). On the other hand, the vulnerability factors that received particularly few mentions were Gender (0), Access to technology (0), Low levels of economic growth (1), and Partial democracy (1).

In the total number of review, vulnerabilities with the highest number of mentions were, from the top, Better governance (8), Dependence on agriculture (6 mentions), Dependence on and access to natural resources (5), and High population (5). On the other hand, the vulnerability factors that received particularly few mentions were Land degradation/land cover (0), Geographical conditions (0), Customs (2), and Gender (3).

In the total number of mixed studies, vulnerabilities with the highest number of mentions were, from the top, Dependence on agriculture (4), Low level of development (4), Customs (3), High population (3), and Reduced political and ethnic marginalization (3). On the other hand, vulnerability factors that received particularly few mentions were Gender (0), Geographical conditions (0), Land degradation/land cover (1), Development of the economic system (1), Access to technology (1), and Low levels of economic growth (1).

### 3.2. Sensitivity

3.2.1. Dependence on and Access to Natural Resources

Dependence on and access to natural resources was mentioned in a total of 23 studies: 10 out of the 24 quantitative studies, 5 out of the 13 qualitative studies, 5 out of the

12 reviews, and 3 out of the 4 mixed-methods studies (Figure 2). Dependence on and access to natural resources, such as freshwater, greatly affects a region's sensitivity to climate change.

Droughts and other effects reduce residents' access to drinking water and other resources. The studies included in this review noted that these impacts increase vulnerability to climate change, increasing the likelihood of community conflict and other issues [28,29]. For example, a study of the southeastern coast of Lake Chad noted that one of the reasons for approximately 75% of the conflicts reported in rural areas was a lack of resources [30]. In this region, lakes play a central role in the livelihoods of farmers [30]. The relatively high dependence of rural villages on lakes contributed to changes in farmers' incomes when lake levels were low or water quality deteriorated [30]. Indeed, it has also been noted that development programs for village support for water supply can be an appropriate intervention for local people, especially pastoralists, who need safe water. If such interventions are locally determined and centrally implemented, they can minimize aggression that frequently occurs among resource users during periods of water scarcity [31].

However, it has also been noted that in sub-Saharan Africa, higher rainfall increases the likelihood of communal conflict [32]. For example, the likelihood of conflict may increase owing to the effects of flooding and raiding of livestock during the rainy season [32].

In addition, people engaged in agriculture depend to a large extent on access to natural resources such as land and water, as well as technical inputs such as facilities, training, information, seeds, tools, fertilizers, and pesticides related to agriculture in order to earn a profit [33]; thus, it is noted that interactions with other vulnerability factors increase vulnerability.

However, it has also been noted that the abundance of certain types of resources increases the probability of conflict. In particular, the extraction and export of resources (especially oil) increases the (occurrence) probability of conflict because rent-seeking behavior further increases the financial incentive to engage in conflict, and the nation-state and mining companies have more benefits than local unskilled labor [34].

### 3.2.2. Dependence on Agriculture

Dependence on agriculture was mentioned in a total of 19 studies; 7 out of the 24 quantitative studies, 2 out of the 13 qualitative studies, 6 out of the 12 reviews, and 4 out of the 4 mixed-methods studies (Figure 2).

Agriculture is sensitive to short-term shocks from extreme weather events and long-term climate change. In particular, rain-fed agriculture, which relies on rainwater, is more strongly affected by adverse weather conditions than irrigated agriculture [35]. Many studies have indicated that the effects of drought increase the risk of conflict, especially in areas that rely on rainfed agriculture [35]. For example, areas in Syria that depended on irrigated agriculture showed little increase in vulnerability after subsidies for agriculture were reduced [36]. Instead, it was noted that vulnerability was somewhat higher in areas dependent on both irrigated and rainfed agriculture, and in areas with less than moderate levels of irrigation [36]. Moreover, the Punjab province of Pakistan, for instance, is poor not only because many rural livelihoods remain heavily dependent on agriculture but also because of limited access to productive assets such as land, labor, fertilizer, infrastructure, and financial services [37]. Many rural areas suffer from severe poverty and limited access to agricultural resources; in these areas, the amount of crops that can be harvested is lower than the amount that can potentially be harvested [37]. Poor households typically do not have access to improved seeds, advanced technologies, or other inputs to reduce crop vulnerability to climate-related risks. As a result, small and poor farmers have little capacity to adapt to climate change. Even a small loss of income can be devastating, and the lack of limited assets and absence of economic and social safety nets can lead to further poverty and future vulnerability [37]. At the same time, a study conducted in Zambia noted that only farmers who have their agricultural water supplies filled have time to participate in political and violent conflicts [35].

### 3.2.3. Land Degradation/Land Cover

Land degradation/land cover was mentioned in a total of 7 studies; 3 out of the 24 quantitative studies, 3 out of the 13 qualitative studies, 0 out of the 12 reviews, and 1 out of the 4 mixed-methods studies (Figure 2).

Existing studies have indicated that conditions under which land degradation degrades the living conditions of people who depend on ecosystems for food and feed are highly vulnerable to climate change [38–40]. Land degradation can be caused by several factors. For example, in Afghanistan, the combined negative effects of conflict, drought, and lack of sustainable land management have been noted to have a significant impact on desertification and land degradation [39]. Furthermore, the increased demand for land caused by rapid population growth and the massive influx of returning refugees has exacerbated desertification and land degradation [39]. In addition, Pakistan has been investigated as a case of illegal loggers causing land degradation by extracting timber to maintain the Taliban arsenal [38]. Land degradation is caused by a variety of factors resulting from industrialization and past conflicts.

Again in Pakistan, extensive deforestation caused landslides during the 2010 flood, resulting in extensive damage [38]. Several cases of deforestation, which increases the impact of flooding, have been reported [41,42].

### 3.2.4. Gender

In recent years, gender has been identified as a key factor in considering vulnerability to climate change. Gender was mentioned in a total of 3 studies: 0 out of the 24 quantitative studies, 0 out of the 13 qualitative studies, 3 out of the 12 reviews, and 0 out of the 4 mixed-methods studies (Figure 2).

For example, in some cases, men migrate out of the country to make a living in response to climate stress; it has been noted that women left behind are at increased risk because of this phenomenon, experiencing heavier work burdens, increased violence, and a threat of trafficking as an indirect result of climate stress. Moreover, gendered roles and responsibilities often link women to the environment through their dependence on natural resources to sustain their livelihoods [43–45].

The link between gender and resource scarcity inevitably places these women in vulnerable positions. Studies in the Philippines, Iran, Afghanistan, and the Himalayas point out that women experience food insecurity, increased workloads, and loss of income during climate stress, which are exacerbated by the prevalence of armed conflict. Furthermore, studies on women's vulnerability in conflict zones indicate that a lack of access to financial resources, land ownership, and health services exacerbates forced displacement and gender-based violence [43–47].

### 3.3. Adaptive Capacity

### 3.3.1. Better Governance

Better governance was mentioned in a total of 33 studies: 13 out of the 24 quantitative studies, 10 out of the 13 qualitative studies, 8 out of the 12 reviews, and 2 out of the 4 mixed-methods studies (Figure 2). Poor governance is known to be one of the factors that increase vulnerability to climate change. For example, cyclone mortality has been shown to be more severe in areas with weak political institutions and an inadequate provision of public goods [48]. Thus, in general, better governance is considered to allow for the provision of infrastructure and other services to protect the rights and freedom of individuals, including minority groups, from the adverse effects of disasters and climate change.

Previous studies have indicated that governance over the allocation of natural resources, such as water, affects vulnerability to climate change. That is, the inequitable distribution of water resources increases the risk of grievances, conflicts, and tensions among people who do not have access to those resources [33,49,50]. For example, it has been noted that the greater the number of rules set by the official government, the less the recourse to violence [51]. On the other hand, it has been noted that the absence of

government rules for natural resource management also decreases the likelihood of conflict [51]. For example, the case of Tanzania points out that the rules governing water contain elements that prevent conflict resolution [52]. It has been noted that the relationship between the number of rules and conflict may be that the rules do not reflect the will of voters well and that new rules may not be prevalent [51].

It has also been noted that trust in state leaders by the public may reduce the level of support for political violence. For example, a survey conducted just four months after the re-election of popular Ghanaian President John Agyekum Kufuor in December 2004 showed that Kufuor, who had implemented ambitious social and economic reforms, advocated adaptation to climate change and protected vulnerable populations from the effects of extreme weather. It has been noted that the level of political violence in Ghana was generally low during that period, despite the fact that many people were affected by the drought [53].

### 3.3.2. Development of Economic System

The development of the economic system, such as access to markets and insurance, was mentioned in a total of 4 studies: 0 out of the 24 quantitative studies, 2 out of the 13 qualitative studies, 1 out of the 12 reviews, and 1 out of the 4 mixed-methods studies (Figure 2).

Primarily, the qualitative research points out that the development of an economic system is related to its ability to adapt to climate change, such as access to markets to sell products [54], insurance schemes against environmental risks [14], and basic access to financial services such as microfinance [55]. For example, countries in South America and the Congo River Basin are at risk of low-level water conflict [28], and such conflicts are likely to escalate as inequality increases [28]. On the other hand, it has also been noted that the risk of conflict eventually decreases given economic development, the opening of trade, and an increase in financial institutions [56]. In addition, access to unequal markets [57] and a lack of financial means [33] also contribute to constraints that hinder adaptation to climate change. For example, according to Abid et al., most farm households in Punjab, Pakistan, report the importance of access to agricultural credit services for climate change adaptation [37]. However, they noted that farmers were reluctant to access such services for reasons, such as high interest rates [37].

### 3.3.3. Access to Technology

Access to technology was mentioned in one mixed-methods study (Figure 2).

It has been noted that access to technology, such as easy access to better equipment and materials, and the provision of information services, play an important role in reducing susceptibility to adverse climate-related impacts [35]. In particular, for farmers, access to climate change-adapted agricultural facilities, training, information, and technical resources such as seeds, tools, fertilizers, and pesticides greatly determines their adaptive capacity [33].

### 3.3.4. Customs

Customs were mentioned in a total of 8 studies; 1 out of the 24 quantitative studies, 2 out of the 13 qualitative studies, 2 out of the 12 reviews, and 3 out of the 4 mixed-methods studies (Figure 2). Customary factors determine the adaptability of each society to climate change. Customs are informal institutions such as traditional norms and practices.

For example, membership in a group such as a village-level organization affects the way local residents bond with each other [58]. Such social-relational capital is crucial for reducing vulnerability to climate change and conflict [35]. In Gujrat, Pakistan, many conflicts between farmers over groundwater use and rainwater harvesting have been reported. Thus, it can be concluded that farmers who cooperate with other farmers are better able to cope with the adverse effects of climate change [37]. In other words, farmers

who are in conflict with other farmers and farms in isolation are less likely to adapt to climate-related risks.

In some areas, the ability to cooperate depends on the industry in which inhabitants are engaged [30]. Fishermen have good access to cooperation and information, and some farmers form agricultural cooperatives [30]. On the other hand, pastoralists have limited socio-political networks [30]. The mobile lives of pastoralists affect their social and political participation and their relationship with authorities at the village and district levels [59]. However, despite occasional visits by agencies providing socioeconomic support, pastoralists reported that such visits have not yet led to strong relationships among villages, agencies, and organizations [30].

### 3.3.5. Low Level of Development

A low level of development was mentioned in a total of 24 studies; 11 out of the 24 quantitative studies, 3 out of the 13 qualitative studies, 6 out of the 12 reviews, and 4 out of the 4 mixed-methods studies (Figure 2).

Studies included in this review pointed out that low levels of development, such as low access to healthcare and to commodities related to food and water consumption and low levels of education, reduce the ability to adapt to climate change [14]. Furthermore, it has been noted that physical infrastructure plays an important role in reducing susceptibility to adverse climate-related impacts [37]. Thus, low levels of economic and social development depend on the development of social services and infrastructure.

Economic and social development here comprises two main components: social services and infrastructure development [60]. Bretthauer noted that high levels of tertiary education combined with other socioeconomic factors, such as low levels of poverty, diverse livelihood strategies, lack of ethnic divisions contribute to conflict minimization [60]. As the level of social services, such as education, increases, children will be employed in industries that are less vulnerable to the effects of climate change, which will have a significant bearing on their ability to cope with its effects [60].

With regard to infrastructure development, Detges noted that access to key infrastructure to address drought and prevent violence mitigates the impact of drought on the risk of conflict outbreaks. However, the type of infrastructure changes the effect on conflicts. Access to paved roads affects the risk of precipitation shortage–related civil war outbreaks, but does not influence the risk of drought-related mass violence. On the other hand, in areas with inadequate water infrastructure, the results show an effect on the risk of drought-related communal conflict outbreaks but not on the risk of civil war outbreaks. [53]

The reason drought-related conflicts occur in areas with low paved road coverage may be due to topographical features such as rough or inaccessible terrain that facilitate guerrilla warfare [61]. Similarly, drought-related local conflicts occur more frequently in areas with inadequate water infrastructure [62–64]. This may be due to the generally weak performance of resource management institutions in these regions [62–64]. It has also been noted that farmers in Pakistan who rely on rain-fed agriculture are more susceptible to climate change and climate sensitivity than other farmers in Pakistan because of the lack of basic infrastructure for adaptation to adverse climate-related impacts [37].

### 3.4. General Risks for Violent Conflict
### 3.4.1. Low Levels of Economic Growth

Low economic growth was mentioned in a total of 19 studies; 15 out of the 24 quantitative studies, 1 out of the 13 qualitative studies, 2 out of the 12 reviews, and 1 out of the 4 mixed-methods studies (Figure 2).

Among quantitative studies, it was noted that a low per capita GDP [65], in particular, can lead to conflict. For example, in areas where individuals have low economic power, food price spikes can threaten the livelihood of individuals because they cannot afford to purchase groceries [66]. In areas of low vulnerability, a price increase had a 9% predicted probability of violence [66], while in areas of high vulnerability, a similar price increase has

a 44% chance of violence [66]. However, it has also been noted that GDP per capita is a less important risk factor for causing conflict [67].

The GPD per capita is also closely related to food supply. For example, a statistically significant relationship exists between GDP per capita and conflict. However, it has been noted that GDP is less likely to act on conflict when the food supply is high [68].

### 3.4.2. Partial Democracy

Partial democracy was mentioned in a total of 11 studies: 5 out of the 24 quantitative studies, 1 out of the 13 qualitative studies, 3 out of the 12 reviews, and 2 out of the 4 mixed-methods studies (Figure 2).

It has been pointed out that less democratic regions experience more violence [69]. In a democratic system, people can meet the political demands of the government through voting and other means. The government is institutionally responsible for the demands of the people. Disputes between people are mediated by an independent judiciary. Thus, democracies are less likely to experience conflict because they have established processes to protect the rights and freedom of diverse individuals, including minority groups. One of the reasons why democracies are less prone to conflict is that they are more likely to agree to ceasefires and negotiations after major natural disasters [70]. In short, Natural disasters increase the likelihood that parties will initiate talks or agree to ceasefires.

Democracies are less likely to let hunger develop into famine due to effective distributional policies, accountable decision makers, and freedom of the press [71]. These are also associated with superior environmental protection [72]. In vulnerable and undemocratic systems, unsustainable resource management, land use, and discriminatory property rights increase the human security impact of climate change [73–75]. For example, Haiti and the Dominican Republic, neighbors on the Caribbean island of Hispaniola, share many of the same environmental and ecological characteristics [76]. However, the Dominican Republic has had more stable and peaceful political development and socioeconomic growth, and it has excelled in preserving its vegetation as a natural protection against seasonal tropical hurricanes [76].

### 3.4.3. High Population

High population was mentioned in a total of 28 studies; 18 out of the 24 quantitative studies, 2 out of the 13 qualitative studies, 5 out of the 12 reviews, and 3 out of the 4 mixed-methods studies (Figure 2).

It has been noted that population growth is likely to be associated with social unrest [77]. For example, existing studies have indicated that a large population can be difficult for the state to control, that the number of people who may be drawn into armed groups is higher [78], and that governments may have difficulty providing adequate disaster preparedness and post-disaster assistance [79]. In Kenya, it has also been noted that population growth can lead to grievances, especially among youth from rival tribes and with different economic backgrounds (e.g., smallholder farmers, pastoralists, fishermen, etc.). Furthermore, according to Kahl, population growth factors, when combined with governance issues, can lead to state bankruptcy [80].

### 3.4.4. Recent Conflict and Tensions

Recent conflicts and tensions were mentioned in a total of 20 studies; 11 out of the 24 quantitative studies, 3 out of the 13 qualitative studies, 4 out of the 12 reviews, and 2 out of the 4 mixed-methods studies (Figure 2).

In areas with a history of conflict, conflict and violence are more likely to occur [34]. In Guatemala, decades of civil war have left a legacy of violence and unemployment [81]. This event became a hotbed of organized crime by non-state armed groups (NSAGs) and others [81]. Civil wars make people more vulnerable to the negative effects of climate change owing to reduced rural development and environmental degradation. This has forced some people to engage in illegal activities or to migrate to urban areas [81]. In

addition, decades of civil war and other conflicts make it easier to obtain weapons in the area, leading to constant instability [82,83].

### 3.4.5. Reduce Political and Ethnic Marginalization

Reducing political and ethnic marginalization was mentioned in a total of 19 studies; 9 out of the 24 quantitative studies, 3 out of the 13 qualitative studies, 4 out of the 12 reviews, and 3 out of the 4 mixed-methods studies (Figure 2).

The studies included in this review point out that ethnic discrimination and political marginalization lead to conflict and riots [50,84]. For example, those who are politically frustrated with marginalization have more difficulty coping with droughts. They were also more likely to blame the government. Consequently, it has been noted that more radical attitudes may be supported or even lead to violence against the government [53]. Ethnic divisions have also been identified as factors that increase conflict [85]. On the other hand, division by religion may not be statistically significant [85]. Thus, it has been pointed out that ethnic division and political marginalization, which assign superiority or inferiority to ethnic groups based on cultural and religious differences, is one of the main factors that lead to conflict and tension [50].

It has been noted that societies where ethnic exclusion does not exist are nearly 25% less likely to experience conflict due to the effects of drought than societies where ethnic exclusion does exist [68]. This result indicates that, as ethnic exclusion increases, drought becomes less influential as a factor leading to conflict.

### 3.4.6. Geographical Conditions

Geographical conditions were mentioned in a total of 8 studies: 8 out of the 24 quantitative studies, 0 out of the 13 qualitative studies, 0 out of the 12 reviews, and 0 out of the 4 mixed-methods studies (Figure 2).

For example, a quantitative study in sub-Saharan Africa found that border areas are 1.2 times more conflict-prone than other areas [34]. In addition, rugged terrain makes it easier to hide when attacked and risks 1.05 times more fighting than areas with less rugged terrain [86]. Countries that share many international rivers with their neighbors are at a higher risk of water conflict [28]. As like these examples, countries with rougher terrain have been shown to be more conflict-prone because inaccessible areas offer nice hiding place to insurgents and inhibit a state's efforts to reach isolated areas [68].

On the other hand, among the geography variables, rough terrain and noncontiguous territory do not produce a statistically significant impact on the outcome across most model specifications. This finding is consistent with Buhaug et al. (2009), who apply geographic information systems (GIS) techniques to improve the measurement of mountainous and forested terrain in conflict zones [87,88].

## 4. Discussion

As noted above, many previous studies have indicated that whether the effects of climate change escalate to the outbreak of conflict depends, to a large extent, on the different vulnerabilities of each society (sensitivity, adaptive capacity, and general risks for violent conflict). Even when faced with the same level of extreme weather and natural disasters, societies with low vulnerability are less likely to experience climate conflicts and vice versa. This section proposes a vulnerability analysis framework and presents implications for future research based on the findings of previous studies.

### 4.1. Vulnerability Analysis Framework

The classification proposed by Pearson and Newman (2019) is instructive when constructing a framework for analyzing the vulnerability linking climate change and conflict. However, the results of this review suggest two problems with blindly adopting these classifications.

First, the existing framework of Pearson and Newman does not cover the vulnerability factors pointed out in previous studies. For example, the conventional vulnerability framework of Pearson and Newman does not include Customs, Geographical conditions, Land degradation/land cover, and Gender. The number of mentions of Land degradation/land cover, Customs and Geographical conditions, and Gender in previous studies is not significantly different from the number of vulnerability factors, such as the Development of the economic system, Partial democracy and Access to technology, as mentioned in the framework by Pearson and Newman. Also, more attention will need to be paid to the importance of infrastructure. As noted in the "Low level of development" section, access to key infrastructure to address natural disasters mitigates the impact of disasters on the risk of conflict outbreaks [53]. In this regard, when infrastructure is destroyed by natural disasters or the associated conflicts, its resilience varies from society to society, so that it is necessary to consider reconstruction prioritization using cost-based resilience for the benefit of the society [89]. In other words, a society's vulnerability to climate security risks depends not only on the level of development of its infrastructure, but also on its resilience when it is destroyed. Therefore, these factors should be included in the vulnerability analysis framework. However, some studies have pointed out that the relationship between climate change and conflict is not statistically significant, even if Geographical conditions are present [85,87,90].

Second, the Pearson and Newman's criteria for classifying each vulnerability factor require further consideration. For example, they consider a Low level of economic and social development to be a factor that promotes conflict. However, considering the existing literature that mentions these factors, a Low level of development should be classified as a factor that influences adaptive capacity. They also classify Reduced ethnic political and ethnic marginalization as factors that affect adaptive capacity. However, previous research has indicated that ethnic fragmentation within groups is highly correlated with conflicts involving low levels of weaponry regardless of the effects of climate change [19]. In other words, ethnic fragmentation and political marginalization, where cultural and religious differences determine the superiority or inferiority between ethnic groups, are factors that promote conflict rather than just one factor that affects the ability to adapt to climate change. Therefore, these two factors should be classified as general risks for violent conflict rather than as adaptive capacity.

As described above, a modified vulnerability framework that overcomes the challenges of the previous vulnerability analysis framework proposed by Pearson and Newman is proposed in this study (Figure 3). The factors in red in Figure 3 show the modifications from the previous framework.

### 4.2. Implications for Future Research

Extreme weather events or natural disasters resulting from climate change may or may not generate violent conflicts in turn in complex causation loops. Conflict is an extreme and infrequent outcome of climate change that only happens under specific circumstances and is not a given. There are conflicting views in academia regarding the causal link between climate change and conflict, despite the fact that a substantial amount of prior research suggests that it may increase the likelihood of violent conflict.

Based on a review of previous studies, this study offers five suggestions for future directions in climate security research. The fifth set of suggestions for future research is the same as those suggested by other authors [1]. Several new suggestions have been made.

First, previous research has been less qualitative than quantitative in its analyses. In the future, it will be necessary to accumulate case studies on the particular circumstances of vulnerability in each country and region, where the effects of climate change cause conflicts. In this review, there were approximately one-half as many qualitative studies as quantitative studies (13 to 24).

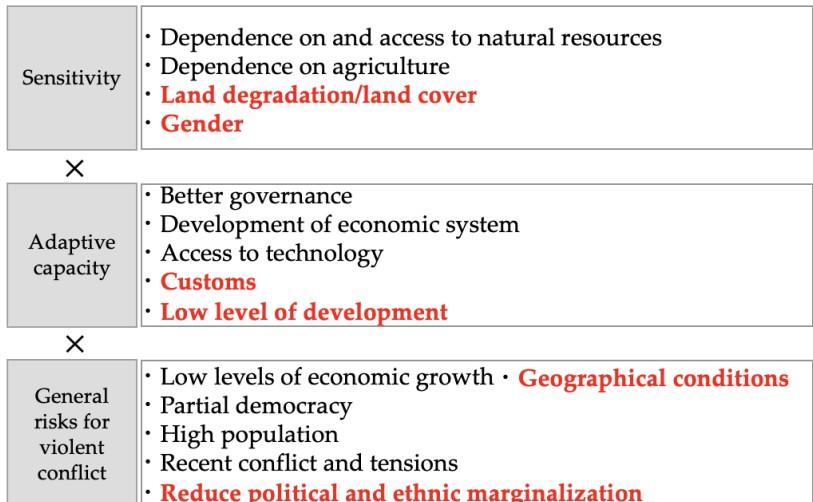

**Figure 3.** Modified vulnerability analysis framework. The factors in red show the modifications from the previous framework proposed by Pearson and Newman.

Second, there were cases where conflict did not occur or cooperation was established despite high vulnerability areas. Further research needs to be conducted on these cases. For example, as noted near the start, drought in the Fertile Crescent was a factor in the social instability at the start of the Syrian Civil War [8,9,90], but the same drought did not lead to conflict in neighboring countries [1].

Third, there is a need for further research on factors that have not been well studied within the conventional framework. For example, this review found that quantitative studies concentrated their analysis on factors such as "High population" and "Recent conflicts and tensions". On the other hand, factors such as "Access to technology" and "Partial democracy" have not been analyzed much. Future research should include vulnerability factors that have not often been considered in previous studies.

Fourth, each vulnerability factor did not lead to conflict in isolation. Vulnerability conditions can be considered to interact with each other, and their connections must be studied. Some case studies included in this review have pointed out various relationships between factors. However, no study has systematically identified which vulnerability factors are likely to lead to conflict when they interact (i.e., horizontal linkages between vulnerability factors). Therefore, future research should study relationships among and combinations of vulnerability factors that are more likely to lead to conflict.

Finally, the results of this review confirm once again that research is concentrated on countries and regions that are experiencing conflict, and that there is a lack of research on countries and regions that are experiencing similar climate-related environmental changes but have not yet entered into conflict. For example, there is little research in East Asia, with very few exceptions [87,91,92]. The concentration of research in countries that have experienced more conflict can lead to a better understanding of the complex relationships between climate-related environmental change and conflict. However, the concentration on countries where it is easier to conduct such research triggers the so-called "street light effect" [93]. To properly understand how climate-related environmental changes and conflicts are linked, it is necessary to accumulate case studies on vulnerability factors in countries and regions that are experiencing the effects of climate change but have not experienced conflict.

## 5. Conclusions

This study identified the factors that influence each society's vulnerability to climate conflict by systematically reviewing 53 quantitative, qualitative, review, and mixed-methods studies that mention vulnerability factors related to climate change and conflict published between 2007 and 2022. The research findings revealed three main points. First,

through the review of 53 existing studies, four conditions were pointed out that were not included in the conventional vulnerability analysis framework: Land degradation/land cover, Gender, Customs, and Geographical conditions. second, there have been few studies on Access to technology (such as climate change adaptation) and Partial democracy in the previous studies. Third, based on the results of this review, it is necessary to revise the classification criteria in the previous vulnerability analysis framework, because these criteria are not always accurate. In addition, this study offered five suggestions for future research directions in climate security research.

The significance of this study is to have improved the ability of researchers and policy makers to understand the variables that need to be considered in assessing future conflict risk associated with climate change and identifying appropriate adaptation strategies. This study proposed a modified vulnerability analysis framework that overcomed the challenges of the conventional vulnerability analysis framework proposed by Pearson and Newman [13]. The framework considers the complex interplay of various factors that mediate the impact of climate change on vulnerable communities and assesses the risk of violent conflict. By identifying the key elements of sensitivity, adaptive capacity, and general risks for violent conflict, the framework can provide a useful lens for researchers and policy maker to assess vulnerability and potential conflict risk associated with climate change. While these characteristics are to some extent common to the existing vulnerability analysis framework, the original contribution of this study is to shed light on land degradation, gender, customs, and geopolitical conditions as vulnerability factors as well.

On the other hand, there are some remaining issues that were not completed in this study. First, it was not within the scope of this study to examine what data could be used to analyze the factors included in the framework proposed by this study. Second, this study did not consider the differences in importance among the 15 factors included in the framework. The importance of each factor is expected to vary from case to case. Nevertheless, the main challenge factors that are common to many societies under a specific set of conditions should be clarified through the accumulation of numerous case studies. These points should be overcome in order to adapt this study's vulnerability analysis framework to future case studies.

**Author Contributions:** T.N.: Conceptualization (supporting); writing—original draft (lead); writing—review and editing (lead). T.S.: Conceptualization (lead); writing—review and editing (supporting). All authors have read and agreed to the published version of the manuscript.

**Funding:** This research received no external funding.

**Institutional Review Board Statement:** Not applicable.

**Informed Consent Statement:** Not applicable.

**Data Availability Statement:** All data that support the findings of this study are included within the article.

**Acknowledgments:** The authors would like to thank the anonymous reviewers for their valuable comments, which have helped to improve this paper considerably.

**Conflicts of Interest:** The authors declare no conflict of interest.

## Appendix A. List of Studies Reviewed

**Table 1.** List of Studies Reviewed (Quantitative).

| | Author | Date | Journal | Title | Country or Region | Methods |
|---|---|---|---|---|---|---|
| 1 | Owain et al. | 2018 | Palgrave Communications | Assessing the relative contribution of economic, political and environmental factors on past conflict on and the displacement of people in East Africa | East Africa | Quantitative |
| 2 | von Uexkull et al. | 2016 | Proceedings of the National Academy of Sciences of the United Stats of America | Civil conflict sensitivity to growing-season drought | Asia and Africa | Quantitative |
| 3 | Cappelli et al. | 2022 | Economia Politica | Climate change and armed conflicts in Africa: temporal persistence, non-linear climate impact and geographical spillovers | Africa | Quantitative |
| 4 | Fjelde et al. | 2012 | Political Geography | Climate triggers: rainfall anomalies, vulnerability and communal conflict in Sub-Saharan Africa | Sub-Saharan Africa | Quantitative |
| 5 | Döring | 2020 | Political Geography | Come rain, or come wells: How access to groundwater affects communal violence | Africa and the Middle East | Quantitative |
| 6 | Bell et al. | 2016 | Foreign Policy Analysis | Conditional relationships between drought and civil conflict in Sub-Saharan Africa | Sub-Saharan Africa | Quantitative |
| 7 | Gizelis et al. | 2021 | Political Geography | Conflict on the urban fringe: urbanization, environmental stress, and urban unrest in Africa | Africa | Quantitative |
| 8 | Cao et al. | 2022 | Defence and Peace Economics | Drought, local public goods, and inter-communal conflicts: testing the mediating effects of public effects service provisions | Africa (9 countries) | Quantitative |
| 9 | Detges | 2017 | Political Geography | Droughts, state-citizen relations and support for political violence in Sub-Saharan Africa: a micro-level Africa: analysis | Sub-Saharan Africa | Quantitative |
| 10 | O'Loughlin et al. | 2014 | PNAS | Effects of temperature and precipitation variability on the risk of violence in sub-Saharan Africa, 1980–2012 | Sub-Saharan Africa | Quantitative |
| 11 | Kim | 2021 | Conflict Management and Peace Science | Environmental shocks, civil conflict and aid effectiveness | low- and middle-income countries | Quantitative |
| 12 | T Jones | 2017 | Journal of Peace Research | Food scarcity and state vulnerability: Unpacking the link between climate variability and violent variability unrest | Africa | Quantitative |
| 13 | Eastin | 2016 | International Interactions | Fuel to the fire: natural disasters and the duration of civil conflict | NA | Quantitative |
| 14 | Bakker et al. | 2017 | Water International | Future bottlenecks in international river basins: where transboundary institutions, population growth and hydrological growth variability intersect | NA | Quantitative |
| 15 | Detges | 2016 | Journal of Peace Research | Local conditions of drought-related violence in sub-Saharan Africa: The role of road and of water infrastructures | Sub-Saharan Africa | Quantitative |

**Table 1.** *Cont.*

| | Author | Date | Journal | Title | Country or Region | Methods |
|---|---|---|---|---|---|---|
| 16 | Hoch et al. | 2021 | Environmental Research Letters | Projecting armed conflict risk in Africa towards 2050 along the SSP-RCP scenarios: a SSP-RCP machine learning approach | Africa | Quantitative |
| 17 | Linke et al. | 2015 | Global Environmental Change | Rainfall variability and violence in rural Kenya: Investigating the effects of drought and of the role of local institutions with survey data | Kenya | Quantitative |
| 18 | Eklund et al. | 2022 | Communications Earth & Environment | Societal drought vulnerability and the Syrian climate-conflict nexus are better explained by agriculture explained than meteorology | Syria | Quantitative |
| 19 | von Uexkull | 2014 | Political Geography | Sustained drought, vulnerability and civil conflict in Sub-Saharan Africa | Sub-Saharan Africa | Quantitative |
| 20 | Breckner et al. | 2019 | World Development | Temperature extremes, global warming, and armed conflict: new insights from high resolution data high | Africa | Quantitative |
| 21 | Gunasekara et al. | 2013 | Water Resour Manage | Water conflict risk due to water resource availability and unequal distribution distribution | NA | Quantitative |
| 22 | Regan et al. | 2019 | Regional Environmental Change | Water scarcity, climate adaptation, and armed conflict: insights from Africa | Africa | Quantitative |
| 23 | Carrão et al. | 2016 | Global Environmental Change | Mapping global patterns of drought risk: an empirical framework based on sub-national estimates of hazard, exposure and vulnerability | NA | Quantitative |
| 24 | Linke et al. | 2018 | Journal of Conflict Resolution | Drought, local institutional contexts, and support for violence in Kenya | Kenya | Quantitative |

**Table 2.** List of Studies Reviewed (Qualitative).

| | Author | Date | Journal | Title | Country | Methods |
|---|---|---|---|---|---|---|
| 1 | Muzamil et al. | 2021 | Regional Environmental Change | An extreme climatic event and systemic vulnerabilities in the face of conflict: insights from the Taliban insurgency in Swat, Pakistan | Pakistan | Quantitative |
| 2 | Tshimanga et al. | 2021 | Sustainability | An integrated information system of climate-water-migrations-conflicts nexus in the Congo Basin | Congo | Quantitative |
| 3 | Cappelli et al. | 2021 | The Journal of Peasant Studies | Climate change as the last trigger in a long-lasting conflict: the production of vulnerability in northern Guinea-Bissau, West Africa | Guinea-Bissau | Quantitative |
| 4 | Heikkinen | 2021 | Regional Environmental Change | Climate change, power, and vulnerabilities in the Peruvian Highlands | Peru | Quantitative |
| 5 | Klein et al. | 2018 | Environment and History | Climate, conflict and society: changing responses to weather extremes in nineteenth century Zululand | Zululand | Quantitative |
| 6 | Khan et al. | 2018 | Climate Policy | Climates of urbanization: local experiences of water security, conflict and cooperation in peri-urban South-Asia | Bangladesh, India, Nepal | Quantitative |
| 7 | Přívara et al. | 2019 | Sustainability | Nexus between climate change, displacement and conflict: Afghanistan case | Afghanistan | Quantitative |
| 8 | Renner et al. | 2019 | ZFW—Advances in Economic Geography | Stakeholders' interactions in managing water resources conflicts: a case of Lake Naivasha, Kenya Lake | Kenya | Quantitative |
| 9 | Schilling et al. | 2015 | Earth System Dynamics | The nexus of oil, conflict, and climate change vulnerability of pastoral communities in northwest Kenya | Kenya | Quantitative |
| 10 | Lynch | 2012 | Global Environmental Change | Vulnerabilities, competition and rights in a context of climate change toward equitable water governance in Peru's Rio Santa Valley | Peru | Quantitative |
| 11 | Sovacool | 2018 | World Development | Bamboo beating bandits: conflict, inequality, and vulnerability in the political ecology of climate change adaptation in Bangladesh | Bangladesh | Quantitative |
| 12 | Chandra et al. | 2017 | Journal of Rural Studies | Gendered vulnerabilities of smallholder farmers to climate change in conflict-prone areas: a case study from Mindanao, Philippines | Philippines | Quantitative |
| 13 | Okpara et al. | 2017 | Regional Environmental Change | Using a novel climate–water conflict vulnerability index to capture double exposures in Lake Chad | Chad | Quantitative |

**Table 3.** List of Studies Reviewed (Review).

| | Author | Date | Journal | Title | Country or Region | Methods |
|---|---|---|---|---|---|---|
| 1 | Augsten et al. | 2022 | Regional Environmental Change | The human dimensions of the climate risk and armed conflict nexus: a review article | Asia and Africa | Review |
| 2 | Damacena | 2021 | Environmental Policy and Law | Climate change, public insecurity and law: conflicts over water resources in the Brazilian context | NA | Review |
| 3 | Gilmore et al. | 2021 | WIREs Climate Change | Climate mitigation policies and the potential pathways to conflict: outlining a research agenda | NA | Review |
| 4 | Sharifi et al. | 2021 | Environmental Research Letters | Climate-induced stressors to peace: a review of recent literature | NA | Review |
| 5 | Zeitoun et al. | 2011 | Climate and Development | Conflict and social vulnerability to climate change: lessons from Gaza | Gaza | Review |
| 6 | Peters et al. | 2020 | International Journal of Disaster Risk Science | Critiquing and joining intersections of disaster, conflict, and peace research | NA | Review |
| 7 | Ide et al. | 2021 | Politics and Governance | Gender in the climate-conflict nexus: forgotten variables, alternative securities, and hidden power dimensions | NA | Review |
| 8 | Raleigh | 2010 | International Studies Review | Political marginalization, climate change, and conflict in African Sahel states | Sub-Saharan Africa | Review |
| 9 | Abrahams et al. | 2017 | Current Climate Change Reports | Understanding the connections between climate change and conflict: contributions from geography and political geography ecology | NA | Review |
| 10 | Buhaug et al. | 2021 | Annual Review of Environment and Resources | Vicious circles: violence, vulnerability, and climate change | NA | Review |
| 11 | Daoudy et al. | 2022 | WIREs Water | What is climate security? Framing risks around water, food, and migration in the Middle East and North Africa | Middle East and North Africa | Review |
| 12 | Tubi et al. | 2019 | Regional Environmental Change | Changing drought vulnerabilities of marginalized resource-dependent groups: a long-term perspective of Israel's Negev of Bedouin | Israel | Review |

**Table 4.** List of Studies Reviewed (Mixed).

| | Author | Date | Journal | Title | Country or Region | Methods |
|---|---|---|---|---|---|---|
| 1 | Ide et al. | 2020 | Global Environmental Change | Multi-method evidence for when and how climate-related disasters contribute to armed conflict risk armed | NA | Mixed |
| 2 | Ide et al. | 2014 | Political Geography | On exposure, vulnerability and violence: spatial distribution of risk factors for climate change for and violent conflict across Kenya and Uganda | Kenya and Uganda | Mixed |
| 3 | Abid et al. | 2016 | Science of the Total Environment | Climate change vulnerability, adaptation and risk perceptions at farm level in Punjab, Pakistan | Punjab, Pakistan | Mixed |
| 4 | Marcantonio et al. | 2018 | Sustainability | Farmer perceptions of conflict related to water in Zambia | Zambia | Mixed |

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
