# Peer review of "Review of Vulnerability Factors Linking Climate Change and Conflict"

_climate, doi:10.3390/cli11050104_

Round 1

Reviewer 1 Report

This is a literature review paper on vulnerability factors that link climate change and conflicts. The study is relevant and timely. The overall structure is satisfactory, the following comments are provided.

- it is suggested to include in the literature review, relevant studies on conflict resilience: https://doi.org/10.1016/j.scs.2023.104405 and related references.

- not clear how the authors selected the 15 indicators for vulnerability, please explain. For example, why critical infrastructure is not included in the list? although "Low level of development" is linked to infrastructure sector, this can be better considered in the vulnerability classification.

- Geographical conditions are particularly relevant to climate change impact, and it is suggested to expand this part

- the proposed framework includes a list of indicators, however, no discussion or guidance is provided on how these indicators may be classified and assessed, what type of data is needed or which are the main challenges. Also, not clear what is the scale of analysis (eg local, regional, national, global?)

- check for typos: from the top are, from the top

Author Response

Dear Editor,
 ã€€Thank you very much for your email dated April 7 regarding my manuscript climate-2311811 with comments from the reviewers. We submit the revised version of our manuscript titled “Review of vulnerability factors linking climate change and conflict.”

    We thank all the reviewers for their helpful comments. In accordance with the advises, the following revisions have been made in the manuscript. All the revisions are marked up using the “Track Changes” function.

  1. it is suggested to include in the literature review, relevant studies on conflict resilience: https://doi.org/10.1016/j.scs.2023.104405 and related references.

We referenced this paper in Line 810-813.

  1. not clear how the authors selected the 15 indicators for vulnerability, please explain. For example, why critical infrastructure is not included in the list? Although “Low level of development” is linked to infrastructure sector, this can be better considered in the vulnerability classification.

We added the explanation of classification in Line 154-169. Also, we added sentences to emphasize the importance of infrastructure resilience referring to the paper above in Line 807-815.

  1. Geographical conditions are particularly relevant to climate change impact, and it is suggested to expand this part.

We made some changes in Geographical conditions (Line 771-778) sections.

  1. the proposed framework includes a list of indicators, however, no discussion or guidance is provided on how these indicators may be classified and assessed, what type of data is needed or which are the main challenges. Also, not clear what is the scale of analysis (eg local, regional, national, global?)

We added the explanation of classification in Line 154-169. Some sentences were added to explain the scale of analysis in Line 927-928. In addition, we included your other comments as the limitations of our study in Line 926-934.

  1. check for typos: from the top are, from the top.

We modified “from the top are” and “from the top” (3.1Overview of previous research on vulnerability: Page 5 line 213).

   We believe the manuscript has been improved satisfactorily and hope that it will be acceptable for publication in Frontiers in Political Science. Thank you very much for your consideration.

Sincerely,

Takato Nagano

Reviewer 2 Report

This manuscript is a quite comprehensive assessment of the review which explores and discussed based on 53 representative studies. It is necessary to enhance the way of preparing the manuscript so that the quality of this manuscript can be improved, including:

§   Introduction: could the authors describe more specifically in what field of the study (agriculture, urban, fisheries etc.) the authors intended to review of vulnerability factors linking climate change and conflict?

§   Material and methods: In Table 1 might be better if describing the literature review based on the field of studies. Simplification based on the appendix

§   Page 4 line 159: change Figure 6 to Figure 4.

§   Page 4 line 158-185: check again the representative number on the variable that shows on Figure 2.

§   Page 5 line 193: check figure 3? And check also fig.3 for the others

§   Results: how the authors classified the parameters under the three criteria, seen Figure 2.

§   Page 12 line 534 Fig.4 or Fig.3?

§   4.2 Implication for further research: discuss about linking climate change and conflict before suggesting further research.

§   Conclusions: suggest rewriting the conclusions. Avoid writing a similar statement as already mentioned in the discussion. Write more on the author's perspective based on research findings.

Author Response

Dear Editor,

Thank you very much for your email dated April 7 regarding my manuscript climate-2311811 with comments from the reviewers. We submit the revised version of our manuscript titled “Review of vulnerability factors linking climate change and conflict.”

We thank all the reviewers for their helpful comments. In accordance with the advises, the following revisions have been made in the manuscript. All the revisions are marked up using the “Track Changes” function.

  1. Introduction: could the authors describe more specifically in what field of the study (agriculture, urban, fisheries etc.) the authors intended to review of vulnerability factors linking climate change and conflict?

We made some changes in introduction (Line101-105) sections.

  1. Material and methods: In Table 1 might be better if describing the literature review based on the field of studies. Simplification based on the appendix.

We added Table 2 in Line 154-169.

  1. Page 4 line 159: change Figure 6 to Figure 4.

We modified Page 5 line 215 from Figure 6 to Figure 2.

  1. Page 4 line 158-185: check again the representative number on the variable that shows on Figure 2.

We modified Page 5 line 217-244 in line with Figure 2.

  1. Page 5 line 193: check figure 3? And check also fig.3 for the others

We modified Page 6 line 446 from Figure 3 to Figure 2.

  1. Results: how the authors classified the parameters under the three criteria, seen Figure 2.

We added the explanation of classification in Materials and Methods section (Line 154-169).

  1. Page 12 line 534 Fig.4 or Fig.3?

We modified Page 13 line 837 from Figure 4 to Figure 3.

  1. 2 Implication for further research: discuss about linking climate change and conflict before suggesting further research.

We added a paragraph regarding the link between climate change and conflicts in Line 840-859.

  1. Conclusions: suggest rewriting the conclusions. Avoid writing a similar statement as already mentioned in the discussion. Write more on the author's perspective based on research findings.

We rewrote the conclusion (Line 899-934).

We believe the manuscript has been improved satisfactorily and hope that it will be acceptable for publication in Frontiers in Political Science. Thank you very much for your consideration.

Sincerely,

Takato Nagano

Round 2

Reviewer 1 Report

The authors have revised the manuscript, the following comments are provided by the reviewer. Also, it is suggested to carefully proofread the manuscript for improving the language and check for typos.

the "conventional vulnerability" framework is mentioned in the abstract, it is not clear what is the conventional and non-conventional framework, please explain

Schilling et al., Abid et al. : reference number is missing

conflict[20, 21][21][22][23], [21] appears two time, check all citations in the manuscript

Figure 2: legend is too small, please improve readability of all figures

the same references are repeated several times within the same subsection, it is suggested to avoid such repetitions, eg in 3.2.4. Gender , [45] appears five times in one paragraphs. Please check the manuscript for similar cases.

388-402: [55] appears four times within few lines

416-418: [71] appears two times in two lines

disasters[73][73]: 73 is duplicate

techniques to improve the measurement of 499 [90]mountainous and forested terrain in conflict zones[90][91]. : [90] appears two times in one sentence

environmental risks[14]Gr[14], check for typos, what is 1 and Gr ? 

development.[63]. : remove . before [63]

"As like these examples, countries with rougher terrain 493 have been shown to be more conflict-prone because inaccessible areas offer safe havens to 494 insurgents and inhibit a state’s efforts to reach isolated areas" not clear, English needs improvement

numerous case studies. [14]These points : not clear in which sentence [14] fits here

Figure 3: not clear why some words are in red colour here

Figure 3 is copied from Pearson and Newman, and four new terms are added here, while the authors claim that this is a new framework. This framework was proposed by Pearson and Newman and modified here by the authors, hence, it is not a new one. Please revise the figure caption and the corresponding texts to avoid misunderstanding.

Author Response

Dear Editor,
 ã€€Thank you very much for your email dated April 7 regarding my manuscript climate-2311811 with comments from the reviewers. We submit the revised version of our manuscript titled “Review of vulnerability factors linking climate change and conflict.”

    We thank all the reviewers for their helpful comments. In accordance with the advises, the following revisions have been made in the manuscript. All the revisions are marked up using the “Track Changes” function.

  1. the "conventional vulnerability" framework is mentioned in the abstract, it is not clear what is the conventional and non-conventional framework, please explain.

We made some changes in abstract(Line 13-17) section.

  1. Schilling et al., Abid et al. : reference number is missing

   We referenced this paper in Line 73-76, 615-617.

  1. conflict[20, 21][21][22][23], [21] appears two time, check all citations in the manuscript

We changed from [20, 21] to [20] [21] in Line 94-95.

  1. Figure 2: legend is too small, please improve readability of all figures

We made the text in Figure 2 a bit bigger for better readability.

  1. the same references are repeated several times within the same subsection, it is suggested to avoid such repetitions, eg in 3.2.4. Gender , [45] appears five times in one paragraphs. Please check the manuscript for similar cases.

We modified it.

  1. 388-402: [55] appears four times within few lines

We modified it (Line 545-557).

  1. 416-418: [71] appears two times in two lines

We modified it (Line 712-714).

  1. disasters[73][73]: 73 is duplicate

We deleted another [73] in Line 725-727.

  1. techniques to improve the measurement of 499 [90]mountainous and forested terrain in conflict zones[90][91]. : [90] appears two times in one sentence

   We deleted [90] in Line 806-810.

  1. environmental risks[14]1 Gr[14], check for typos, what is 1 and Gr ? 

We deleted 1 Gr[14] in Line 608.

  1. [63]. : remove . before [63]

We deleted”. “in Line 666.

  1. "As like these examples, countries with rougher terrain 493 have been shown to be more conflict-prone because inaccessible areas offer safe havens to 494 insurgents and inhibit a state’s efforts to reach isolated areas" not clear, English needs improvement

We made some changes in Line 803-805.

  1. numerous case studies. [14]These points : not clear in which sentence [14] fits here

This[14] is typo. We deleted this[14] in Line1002-1006.

  1. Figure 3: not clear why some words are in red colour here

We added an explanation in Line 918-919.

  1. Figure 3 is copied from Pearson and Newman, and four new terms are added here, while the authors claim that this is a new framework. This framework was proposed by Pearson and Newman and modified here by the authors, hence, it is not a new one. Please revise the figure caption and the corresponding texts to avoid misunderstanding.

We changed “a new framework” into “a modified framework” in Line 17,113, 870, 918, 988.
